# Collateral effect of COVID-19 on orthopedic and trauma surgery

**Thomas M. Randau** [1], **Max Jaenisch** [1], **Henryk Haffer**[2], **Friederike Schömig**[2],
**Adnan Kasapovic**[1], **Katharina Olejniczak**[3], **Johannes Flechtenmacher**[4], **Carsten Perka**[2],
**Dieter C. Wirtz**[1], **Matthias Pumberger** [2]*

1 Department of Orthopedic and Trauma Surgery, University Hospital Bonn, Bonn, Germany, 2 Center for
Musculoskeletal Surgery, Charité – University Medicine Berlin, Berlin, Germany, 3 Center for Evaluation and
Methods, University of Bonn, Bonn, Germany, 4 Professional Association for Orthopedic and Trauma
Surgery, Berlin, Germany

☯ These authors contributed equally to this work.
* matthias.pumberger@charite.de

Zurich, SWITZERLAND

**Data Availability Statement:** All relevant data are
within the manuscript and its Supporting
Information files.

**Funding:** The author(s) received no specific
funding for this work.

## Abstract

### Objectives

The purpose of this study was to assess the impact of the COVID-19 pandemic on orthopedic and trauma surgery in private practices and hospitals in Germany.

### Design

In this cross-sectional study, an online-based anonymous survey was conducted from April 2th to April 16th 2020.

### Setting

The survey was conducted among 15.0000 of 18.000 orthopedic and trauma surgeons in Germany, both in private practices and hospitals.

### Participants

All members of the German Society of Orthopedic and Trauma Surgery (DGOU) and the Professional Association for Orthopedic and Trauma Surgery (BVOU). were invited by e-mail to participate in the survey.

### Main outcome measures

Out of 50 questions 42 were designed to enquire a certain dimension of the pandemic impact and contribute to one of six indices, namely "Preparedness", "Resources", "Reduction", "Informedness", "Concern", and "Depletion". Data was analyzed in multiple stepwise regression, aiming to identify those factors that independently influenced the indices.

### Results

858 orthopedic and trauma surgeons participated in the survey throughout Germany. In the multiple regression analysis, being employed at a hospital was identified as an independent

**Competing interests:** Dr. Perka reports personal fees from Smith&Nephew, personal fees from Link, personal fees from DePuy/Synthes, personal fees from Aesculap, personal fees from Zimmer, outside the submitted work. This does not alter our adherence to PLOS ONE policies on sharing data and materials.

positive predictor in the indices for "Preparedness", "Resources", and "Informedness" and an independent negative predictor regarding "Depletion". Self-employment was found to be an independent positive predictor of the financial index "Depletion". Female surgeons were identified as an independent variable for a higher level of "Concern".

## Conclusions

The study confirms a distinct impact of the COVID-19 pandemic on orthopedic and trauma surgery in Germany. The containment measures are largely considered appropriate despite severe financial constraints. A substantial lack of personal protective equipment (PPE) is reported. The multiple regression analysis shows that self-employed surgeons are more affected by this shortage as well as by the financial consequences than surgeons working in hospitals.

## What are the new findings

The COVID-19 pandemic has a profound impact on orthopedic and trauma surgery as an unrelated specialty. Self-employed surgeons are affected especially by a shortage of PPE and financial consequences.

## How might it impact on clinical practice in the near future

Political and financial support can now be applied more focused to subgroups in the field of orthopedics and trauma surgery with an increased demand for support. A special emphasis should be set on the support of self-employed surgeons which are a more affected by the shortage of PPE and financial consequences than surgeons working in hospitals.

## Introduction

In Wuhan, Hubei Province in the People's Republic of China, in December 2019, cases of viral pneumonia caused by a hitherto unknown pathogen were reported. A novel coronavirus was identified in the affected patients, referred to as severe acute respiratory acute syndrome coronavirus-2 (SARS-CoV-2) [1–3]. The disease caused by SARS-CoV-2 was called coronavirus disease 2019 (COVID-19) [4–6]. SARS-CoV-2 has spread worldwide and WHO declared it a pandemic on March 11[th], 2020 [7, 8]. The first SARS-CoV-2 infection was confirmed in Germany on 27[th] January 2020 [9]. In March 2020, the German government decided to take far-reaching measures to contain the virus [10]. In preparation for the expected increase in COVID-19 patients and the associated severe courses of the disease, drastic restructuring measures have been initiated in the health system.

Germany reported 174,0098 SARS-CoV-2 infections by 14 May, but only 7,861 COVID-19 deaths [11]. Currently, most German hospitals still have ample supplies of necessary equipment and medication, and intensive care unit (ICU) capacities are still abundant. The German government has therefore received some acclaim for their management of the disease. All medical specialties are affected in their routine—orthopedic and trauma departments had to adopt as well [12]. A number of organizational adjustments have been made in order to increase ICU capacity, allocate personal protective equipment (PPE), and personnel in anticipation of a rapid increase in hospitalization rates. In addition, the German government

decided to postpone all elective surgeries, starting from 12[th] March 2020 until further notice, adding to the economic and organizational burden on our profession. Resident surgeons have been allocated to ICUs and emergency departments to aid in the treatment of COVID-19 patients [13]. Orthopedic and trauma surgeons in private practice try to provide the best possible out-patient care, despite the oftentimes occurring lack of PPE and insufficient financial support. Though the German federal government passed a law on 25[th] of March 2020 to lower the economic burden on hospitals and contract physicians in Germany, the true economic impact remains yet to be determined [14].

To define the currently perceived challenges in orthopedic and trauma surgery, both in the hospital and outpatient sector, we have conducted this cross sectional survey among orthopedic and trauma surgeons in Germany, regarding their specific working environment and the perceived impact of the pandemic on their work. This also served the purpose of presenting complex interactions between the government, hospitals, surgeons in private practice, health insurance providers, the Association of Statutory Health Insurance Physicians (ASHIP), and professional associations. This study aims to identify current challenges in different settings in the field of orthopedic and trauma surgery throughout Germany, deducting implications for future crisis management in one of the countries most affected by SARS-CoV-2 worldwide. Specific focus is set on the variable impacts experienced by individual subgroups within our profession and on the different levels on "Preparedness", "Resources", "Reduction", "Informedness", "Concern", and "Depletion".

## Participants and methods

### Participants

In this cross-sectional study, an online-based anonymous voluntary survey was conducted within the German Society of Orthopedic and Trauma Surgery (DGOU) [15] and the Professional Association for Orthopedic and Trauma Surgery (BVOU) [16] from April 2[th] to April 16[th], 2020 reaching over 15,000 of a total of 18,000 orthopedic and trauma surgeons in Germany [17].

### Survey

The study protocol was reviewed and approved by the institutional ethics board ((Ethics Committee of the Medical Faculty of the University of Bonn, Approval No. #20/127) as an anonymous online survey study. For the study, we designed a questionnaire in German language, containing a total of 50 items, grouped into 10 blocks; the questionnaire, as well as an English translation, are appended as supplementary material. The first 42 questions were designed to query certain dimensions of the pandemic impact, most of them contributing to one of six indices, namely "Preparedness", "Resources", "Reduction", "Informedness", "Concern", and "Depletion", as defined in supplementary material. These questions were defined "index questions". Within the survey, these questions were grouped thematically into blocks and both negative and positive wording was used. Questions regarding the different dimensions were mixed and usually spread over at least two blocks. The first two blocks with a total of 14 questions allowed "does apply", "does not apply", and "neutral/unsure" as answer and were mainly designed to enquire which protective measures had already been taken in the participant's institution. Block 3 asked for the level of reduction in in-patient and out-patient care (elective/ urgent cases), on a 5-degree scale in percent. Blocks 4 to 7 enquired the participants' level of agreement towards statements regarding preparations, handling, medical, and financial consequences of the pandemic and support by the orthopedic associations and the insurances. As answers, a five-point Likert scale was employed, consisting of "fully agree", "rather agree",

"neutral", "rather disagree", and "fully disagree". The last three blocks consisted of "profile questions", multiple-choice questions regarding the professional and personal profile of the participant, including field of employment, speciality, position, size of the unit, affiliation, age, and gender, as well as an open text field for questions and comments directed at the professional associations in the end.

## Data management

Data was exported to SPSS (v. 26, IBM Corporation, Armonk, USA) and cleared of all incomplete data sets; Data from emailed or mailed questionnaires were added manually to the data file, which was then analyzed in SPSS or exported for analysis to GraphPad Prism 8.2.1 (GraphPad Software, San Diego, CA, USA) and STATA v 16 (StataCorp, College Station, TX, USA). To calculate the predefined indices, we normalized the answers of the applicable questions to a scale of 0 to 1, inverting negatives or positives to adjust the direction of questions within each index. The average of all answers within one index generated the final index result. Items regarding the participants' profile (blocks 7 to 10) were coded as dichotomous items, only allowing yes (1) or no (0) as valid answer, resulting in a total of 22 dichotomous "profile" values for each participant.

## Statistical analysis

Quality control of the data was performed by checking for Skewness and Kurtosis for normality in questions on five-point answer scale as well as heteroscedasticity for the calculated indices. We performed descriptive analysis, calculating mean, standard deviation, standard error, and 95% confidence interval where applicable. We conducted a correlation matrix analysis calculating Spearman's R, first among the index questions and the result of the calculated indices, then with the profile questions against each other and the indices (all in GraphPad Prism). Additionally, we performed a factor analysis including Kaiser-Meyer-Olkin's criteria (KMO) for sampling adequacy, to confirm the validity of our indices, and tested for co-linearity among the profile variables (both SPSS). Then we ran a bivariate analysis of multiple Mann-Whitney-U-tests for each dichotome profile variable against the six indices for significant differences between groups having "yes" (1) or "no" (0) in that profile variable and its effect size on each of the six indices. Last, data was analyzed in multiple stepwise regression, eliminating factors that missed significance from the predictive regression model, aiming to find causative rather than coincident correlations.

## Results

The online survey was opened a total of 1785 times, of which 841 entries were complete. Another 17 surveys were sent via mail or email and added manually to the survey, giving a total N of 858 participants. Sample size is therefore regarded sufficient. We saw no Skewness of the data above 1 or below -1, though Kurtosis was low in almost all items. We regarded the data quality as good and the sample to be representative. Table 1 summarizes the data quality control. The Likert skales were transformed to numerical values (range 0 to 1, interval 0.25) to calculate mean, standard deviation (SD) and standard error (SEM) with confidence interval. e.g., "Feels well informed" has a mean of 0.7, meaning participants on average answered shortly below "rather agree (0.75)" on the Likert skale, with a SD of 0.23 (approximately one item up and down, 0.25 points each).

Data was homoscedastic and variable interference of the profile values was low with a mean Variance Inflation Factor (VIF) of 1.63. Next, we conducted a descriptive analysis of all profile questions and index questions, summarizing the results in a narrative fashion. Figs 1 and 2

**Table 1. Data summary of descriptive analysis.**

| | N | mean | SD | SEM | 95% CI | | Skewness | Kurtosis |
|---|---|---|---|---|---|---|---|---|
| **Preparedness** | **856** | **0·5631** | **0·2386** | **0·0082** | **0·5471** | **0·5791** | **-0·1400** | **-0·5331** |
| **Resources** | **846** | **0·4777** | **0·3359** | **0·0116** | **0·4551** | **0·5004** | **0·2488** | **-1·0710** |
| **Reduction** | **856** | **0·6748** | **0·1608** | **0·0055** | **0·6641** | **0·6856** | **-0·6112** | **0·2283** |
| Reduction in outpatient clinic | 851 | 0·7673 | 0·2198 | 0·0075 | 0·7525 | 0·7821 | -0·6279 | -0·3753 |
| Reduction in elective surgery | 782 | 0·8419 | 0·2621 | 0·0094 | 0·8235 | 0·8603 | -1·4970 | 0·8375 |
| Appointment cancellations (outpatient) | 841 | 0·5432 | 0·1853 | 0·0064 | 0·5306 | 0·5557 | 0·1606 | -0·2710 |
| Appointment cancellations (surgery) | 761 | 0·4809 | 0·2399 | 0·0087 | 0·4639 | 0·4980 | 0·5306 | -0·6233 |
| Total patient reduction | 853 | 0·6828 | 0·1868 | 0·0064 | 0·6702 | 0·6953 | -0·2944 | -0·1471 |
| **Informedness** | **856** | **0·5198** | **0·1649** | **0·0056** | **0·5087** | **0·5308** | **-0·1638** | **0·0519** |
| Feels well informed? | 855 | 0·7009 | 0·2382 | 0·0081 | 0·6849 | 0·7169 | -0·8047 | 0·3822 |
| Cooperative network? | 852 | 0·5176 | 0·2533 | 0·0087 | 0·5006 | 0·5346 | 0·0488 | -0·3437 |
| Quality of prof. associaions work? | 828 | 0·6395 | 0·2284 | 0·0079 | 0·6239 | 0·6551 | -0·2817 | 0·0070 |
| Communication insurances? | 802 | 0·3716 | 0·2260 | 0·0080 | 0·3559 | 0·3872 | 0·0156 | -0·3256 |
| Communication ASHIP? | 797 | 0·4733 | 0·2534 | 0·0090 | 0·4557 | 0·4910 | -0·0826 | -0·4676 |
| Support by ASHIP? | 778 | 0·3737 | 0·2655 | 0·0095 | 0·3550 | 0·3924 | 0·1977 | -0·5845 |
| **Concern** | **856** | **0·4948** | **0·1385** | **0·0047** | **0·4855** | **0·5041** | **0·2409** | **0·2783** |
| Healthcare System well prepared? | 856 | 0·4574 | 0·2888 | 0·0099 | 0·4380 | 0·4767 | 0·3635 | -0·8705 |
| Measures necessary? | 804 | 0·8414 | 0·2024 | 0·0071 | 0·8274 | 0·8554 | -1·4930 | 3·0070 |
| Measures sufficient? | 855 | 0·3202 | 0·2544 | 0·0087 | 0·3031 | 0·3373 | 0·7394 | -0·0051 |
| Feels appreciation? | 855 | 0·3693 | 0·2844 | 0·0097 | 0·3502 | 0·3884 | 0·5155 | -0·4855 |
| Return to normal 2020? | 854 | 0·4438 | 0·2623 | 0·0090 | 0·4262 | 0·4614 | 0·3062 | -0·7946 |
| Expect to work outside specialty? | 659 | 0·5842 | 0·2942 | 0·0115 | 0·5617 | 0·6067 | -0·7110 | -0·1049 |
| **Depletion** | **856** | **0·6431** | **0·2132** | **0·0073** | **0·6288** | **0·6574** | **-0·6394** | **0·2892** |
| Financial measures sufficient? | 855 | 0·5719 | 0·2972 | 0·0102 | 0·5520 | 0·5919 | -0·2056 | -0·8346 |
| More financial assurance? | 762 | 0·7772 | 0·2310 | 0·0084 | 0·7608 | 0·7937 | -0·7384 | 0·1499 |
| Economic difficulties? | 771 | 0·7490 | 0·2115 | 0·0076 | 0·7341 | 0·7640 | -0·7670 | 1·1430 |
| Pandemic threatens existence? | 627 | 0·5311 | 0·3176 | 0·0127 | 0·5062 | 0·5560 | -0·4645 | -0·6901 |

Descriptive analysis and data quality, as assessed by N, mean, standard deviation (SD), standard error of the mean (SEM), 95% confidence interval (95% CI), skewness, and kurtosis of all items. SD, SEM and 95% CI was calculated only for those items measured on the 5-degree scale or Likert skale, as well as for the calculated indices. For dichotome or multiselect items, these calculations are not sensibly possible. Abbreviations: ASHIP: Association of Statutory Health Insurance Physicians.

depict the general personal information and results of the survey as an overview. Descriptive analysis on (a) general impact on clinicians and practice, (b) supply of personal protective equipment, (c) impact on patient care and surgery, (d) assessment of measures taken by the government and influence on society, (e) communication and support of the ASHIP and the health insurance providers, and (f) future prospects are depicted in supplementary material (Fig 1).

## Correlations and regression

We conducted the correlation analysis among the index questions, confirming that all questions within one index correlated among each other positively. We confirmed that the dimensions "Preparedness", "Resources", and "Informedness" correlated positively with each other (positive indices), but negatively with the dimensions "Concern" and "Depletion" (negative indices), while "Reduction" was independent of the other indices. Next, we mapped correlation of the profile questions against each other and against the indices, as depicted in Fig 3.

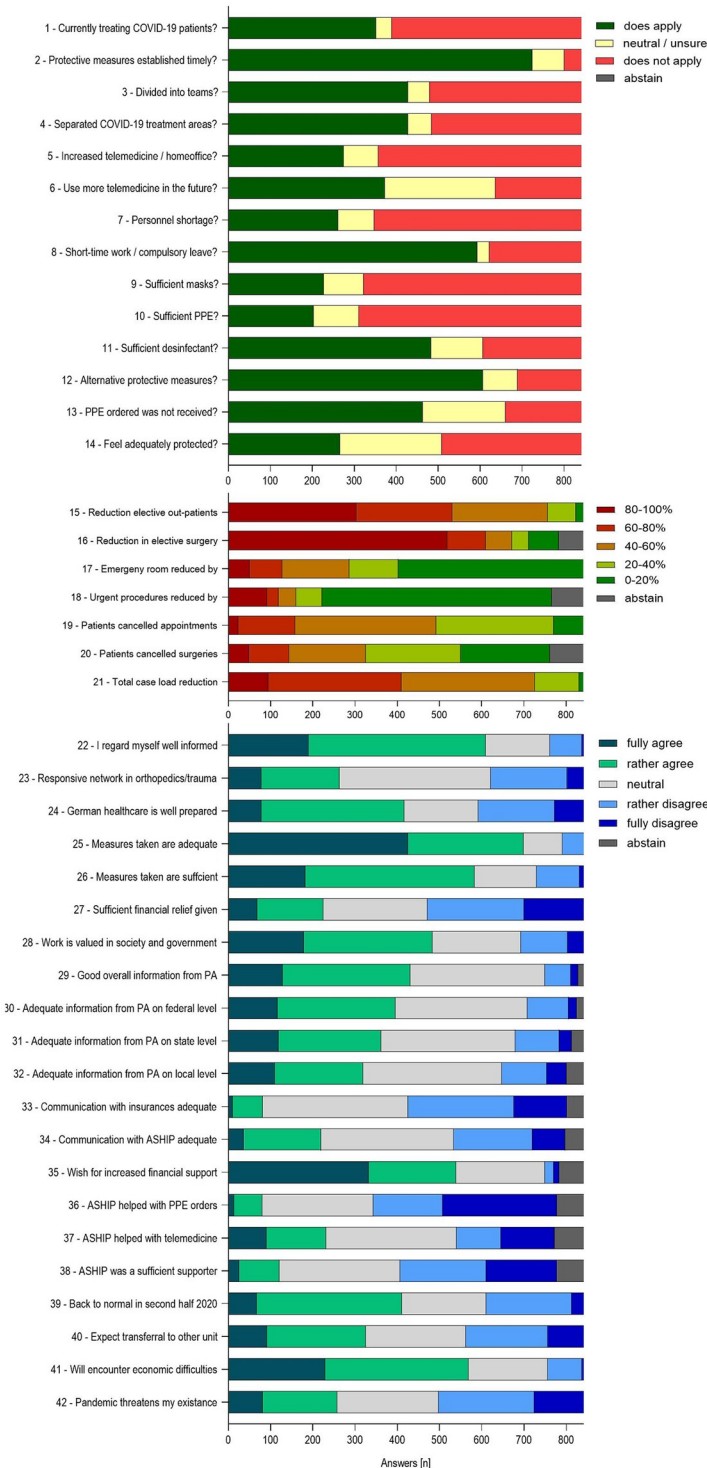

**Fig 1. Overview of the answers to the index questions 1–42 of the questionnaire.** The graph depicts the answers as given by the participants in the 858 fully completed surveys of the first 42 questions, designed to to query certain dimension of the pandemic impact, most of them contributing to one of six indices. For the full question texts, please refer to the questionnaire in the supplemental data. The first 14 questions (Fig 1A) allowed "does apply", "does not apply" and "neutral/unsure" for an answer, 15–21 (Fig 1B) asked for the level of reduction in in-patient and out-patient care, for both elective and urgent cases, on a 5-degree scale in percent. In questions 22 to 42 (Fig 1C), the participants' level of agreement towards statements regarding preparations, handling, medical and financial consequences of the pandemic and support by the orthopaedic associations and the insurances was asked on a five-point Likert scale.

COVID-19: Coronavirus Disease 19; PPE: Personal protection equipment; PA: Professional association of orthopedics and trauma surgeons; ASHIP: Association of statutory health physicians.

Here we saw correlation especially of the parameters indicating surgical specialization rather than conservative against "working in a hospital" and "involved in COVID-19 treatment".

The excellent KMO criteria of 0.830 confirmed the validity of the data set for factor analysis. Though Skewness and Kurtosis were acceptable, the data was formally not normally distributed, likely due to the non-continuous Likert Scale that we employed. We therefore employed Mann-Whitney-U test for the bivariate analysis. Table 2 summarizes the most relevant results of this analysis, comparing the two groups of each possible dependent dichotome profile variable in their outcome regarding each of the six indices as independent variable (6 times 22 U-Tests; with P < 0.05 regarded as significant, the corrected P-Value for

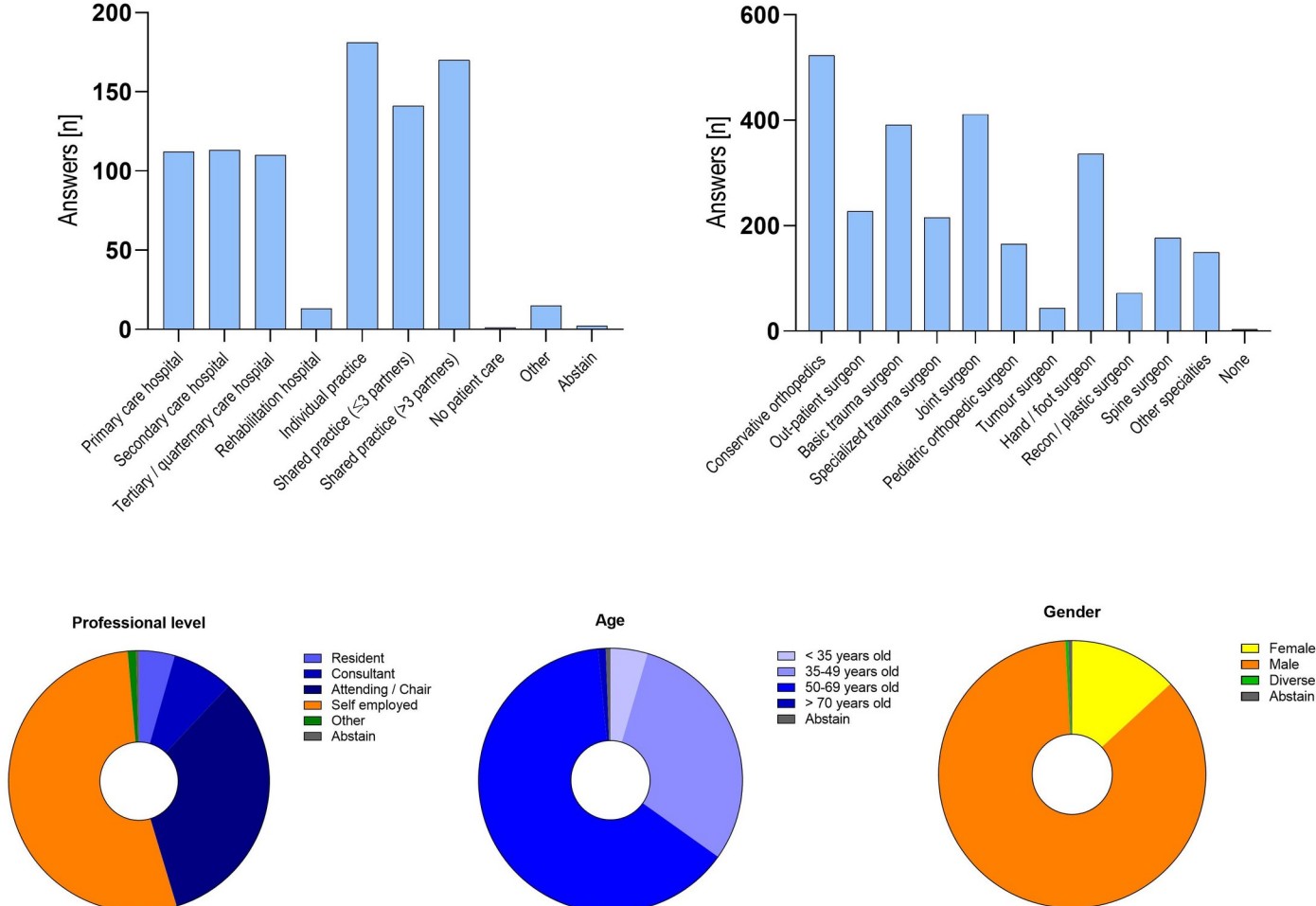

**Fig 2. Overview of the participants' profile.** The graphs show the composition of the participants' profile information of n = 858 fully completed surveys. Fig 2A show the distribution of participants primary work environment; Fig 2B summarizes the sub-specialization of the participants (multiple answers were allowed). Figs 2C, 2D and 2E show the distribution of the participants' professional position, their age group and gender.

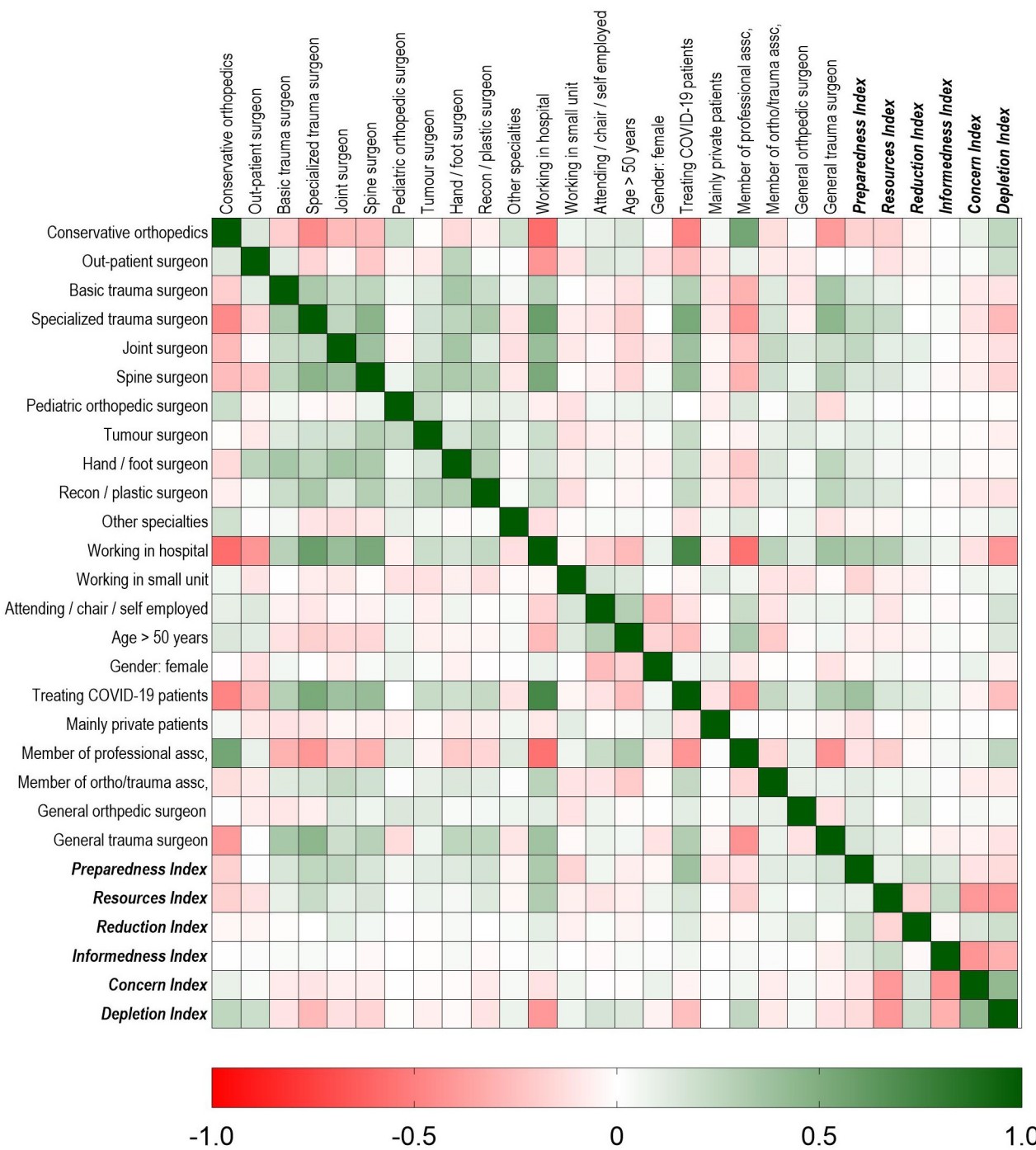

**Fig 3. Correlation matrix of profile items and indices.** The figure shows a heat map of the Spearman correlation between the different profile items and the calculated indices (bold italics, right and bottom). Red boxes indicate a negative correlation (Spearman r < 0), green boxes indicate a positive correlation (Spearman r > 0), with darker color representing stronger correlation and white or light colored boxes no or weak correlation. Surgical specialties, for example, positively correlate with each other, as well as with "Treating COVID-19 patients" and with "Working in hospital". All correlate with higher levels of "Preparedness" and "Resources" indices.

**Table 2. Results of the bivariate analysis.**

| | Effect size r | P -Value | Significant? |
|---|---|---|---|
| Preparedness | | | |
| Treating COVID-19 patients | 0·195 | 0·0000 | Yes |
| Working in hospital | 0·167 | 0·0000 | Yes |
| Plastic / reconstructive surgeon | 0·152 | 0·0000 | Yes |
| Specialized trauma surgeon | 0·146 | 0·0000 | Yes |
| Conservative orthopedics | -0·091 | 0·0000 | Yes |
| Working in a small unit | -0·082 | 0·0000 | Yes |
| Resources | | | |
| Working in hospital | 0·222 | 0·0000 | Yes |
| Specialized trauma surgeon | 0·187 | 0·0000 | Yes |
| Plastic / reconstructive surgeon | 0·170 | 0·0000 | Yes |
| Member of Professional Assoc. (BVOU) | -0·139 | 0·0000 | Yes |
| Conservative orthopedics | -0·138 | 0·0000 | Yes |
| Attending–Chairman–Selfemployed | -0·106 | 0·0050 | No |
| Reduction | | | |
| General orthopedic surgeon | 0·046 | 0·0000 | Yes |
| Treating COVID-19 patients | 0·045 | 0·0010 | No |
| Working in hospital | 0·042 | 0·0030 | No |
| Treating mostly privately insured patients | -0·030 | 0·2770 | No |
| Working in a small unit | -0·023 | 0·1070 | No |
| Conservative orthopedics | -0·019 | 0·2610 | No |
| Informedness | | | |
| Working in Hospital | 0·023 | 0·0910 | No |
| General trauma surgeon | -0·023 | 0·0740 | No |
| Plastic / reconstructive surgeon | -0·022 | 0·4710 | No |
| Spine surgeon | -0·013 | 0·3690 | No |
| Hand / Foot surgeon | 0·013 | 0·3170 | No |
| Basic trauma surgeon | 0·011 | 0·5080 | No |
| Concern | | | |
| Plastic / reconstructive surgeon | -0·042 | 0·0160 | No |
| Gender: Female | 0·031 | 0·0260 | No |
| Specialized trauma surgeon | -0·030 | 0·0030 | No |
| Working in hospital | -0·029 | 0·0010 | No |
| Conservative orthopedics | 0·022 | 0·0140 | No |
| Working in a small unit | 0·015 | 0·1010 | No |
| Depletion | | | |
| Working in hospital | -0·176 | 0·0000 | Yes |
| Specialized trauma surgeon | -0·143 | 0·0000 | Yes |
| Treating COVID-19 P patients | -0·112 | 0·0000 | Yes |
| Conservative orthopedics | 0·110 | 0·0000 | Yes |
| Attending–Chairman–Selfemployed | 0·109 | 0·0000 | Yes |
| Member of Professional Assoc. (BVOU) | 0·107 | 0·0000 | Yes |

Shown are the 6 most significant/most relevant independent profile variables related to the index, as dependent variable, ordered by their effect size. Effect size r estimates the strength of the relationship. Generally, an r > 0·5 is considered a large effect size, 0·1 a small effect size and 0·3 a medium effect size. P values are uncorrected p-values from bivariate analysis. Significance was assumed where P < 0·00038, as conservatively corrected for 132 multiple u-tests.

each test individually was assigned as P < 0.00038, thereby Bonferroni-correcting manually for multiple testing).

The effect size here is an estimate of the strength of the relationship between the profile variable and the indices. Fig 4 illustrates how the indices differ between the subgroups, separated by indices and relevant profile variables (Fig 2).

In regard to "Preparedness" and "Resources" we saw marked positive effects of the profile items "working in a hospital", "treating COVID-19 cases", and being specialized in advanced trauma and reconstructive surgery; participants working in a small unit as non-operative orthopedic physicians had significantly decreased indices. Also, leading consultants saw resources more critically. General orthopedic surgeon is an independent predictor in the index "Reduction". Without reaching significance, colleagues working in a hospital, as well as those specialized in advanced trauma and reconstruction were less concerned, those involved in non-operative treatment, female colleagues and those working in smaller units were more concerned. "Financial depletion" was mostly an issue for colleagues working in non-operative patient care, and for the members of BVOU, while again, specialized traumatologists, hospital doctors, and those involved in treating COVID-19 patients were less concerned about financial losses (Fig 3).

Last, we proceeded to a multiple stepwise regression to identify those factors that independently influence the index rather than just coincide or correlate with the index. For a survey study addressing only a small and defined aspect of the participants' characteristics, fitting of the models was adequate. For "Preparedness", $R^2$ of the model was 0.21, for "Depletion" 0.17, for "Resources", it still reached an $R^2$ of 0.1, the rest remained below 0.05, indicating that the indices were severely influenced by factors that were not queried in our questionnaire. Profile items identified as independent variables are listed in Table 3, together with their incidence rate ratio (IRR), or the delta of IRR to 1 (δIRR), respectively.

The IRR estimates the effect a switch of this variable from 0 ("no") to 1 ("yes") will have on the index in the multiple regression model. Results were somewhat consistent with those from the bivariate analysis, with some added information: For "Preparedness", working in a small unit remained as the only significant negative factor for the model (δIRR: -0.12), while working in a hospital (+0.22) and being involved in COVID-19 patient care (+0.23) were strong positive predictors. Being specialized in joint surgery (+0.08), working as attending /chairman (+0.12), or in outpatient surgery (+0.13) were also weaker independent positive predictors, as was being self-employed (+0.12) or a member of the BVOU (+0.10). "Specialized trauma surgeon" and "conservative orthopedics" did not show up as independent. For "Resources", treating COVID-19 patients was an independent negative predictor (-0.14). "Plastic and Reconstructive Surgery" also showed up as a rather strong positive predictor (+0.19). "Reduction" was only slightly determined by our model at all. It was positively determined by "Specialized in Joint Surgery" (+0.04), "general orthopedic surgeon" (+0.06), and "treating COVID-19 patients" (+0.05) and negatively predicted by "specialized in tumour surgery" (-0.08). Working in a hospital (+0.12) was the strongest predictor for raising the "Informedness" index, as did being 50 years or older (+0.05) and being subspecialized in hand- or foot surgery (+0.05). Working in spine surgery (-0.08) and as general trauma surgeon (-0.08) were weak negative predictors. Female surgeons remained as an independent variable for a slightly higher level of concern (+0.07), working in a hospital reduced concern (-0.05), as did working in general trauma care (-0.04). "Financial depletion" was influenced strongly by the fact "employed in a hospital" (-0.24) and also lowered when specialized as a paediatric orthopedic surgeon (-0.05); Working as attending or chairman, or being self-employed raised this index (+0.12), and so did being a general orthopedic surgeon (Fig 4).

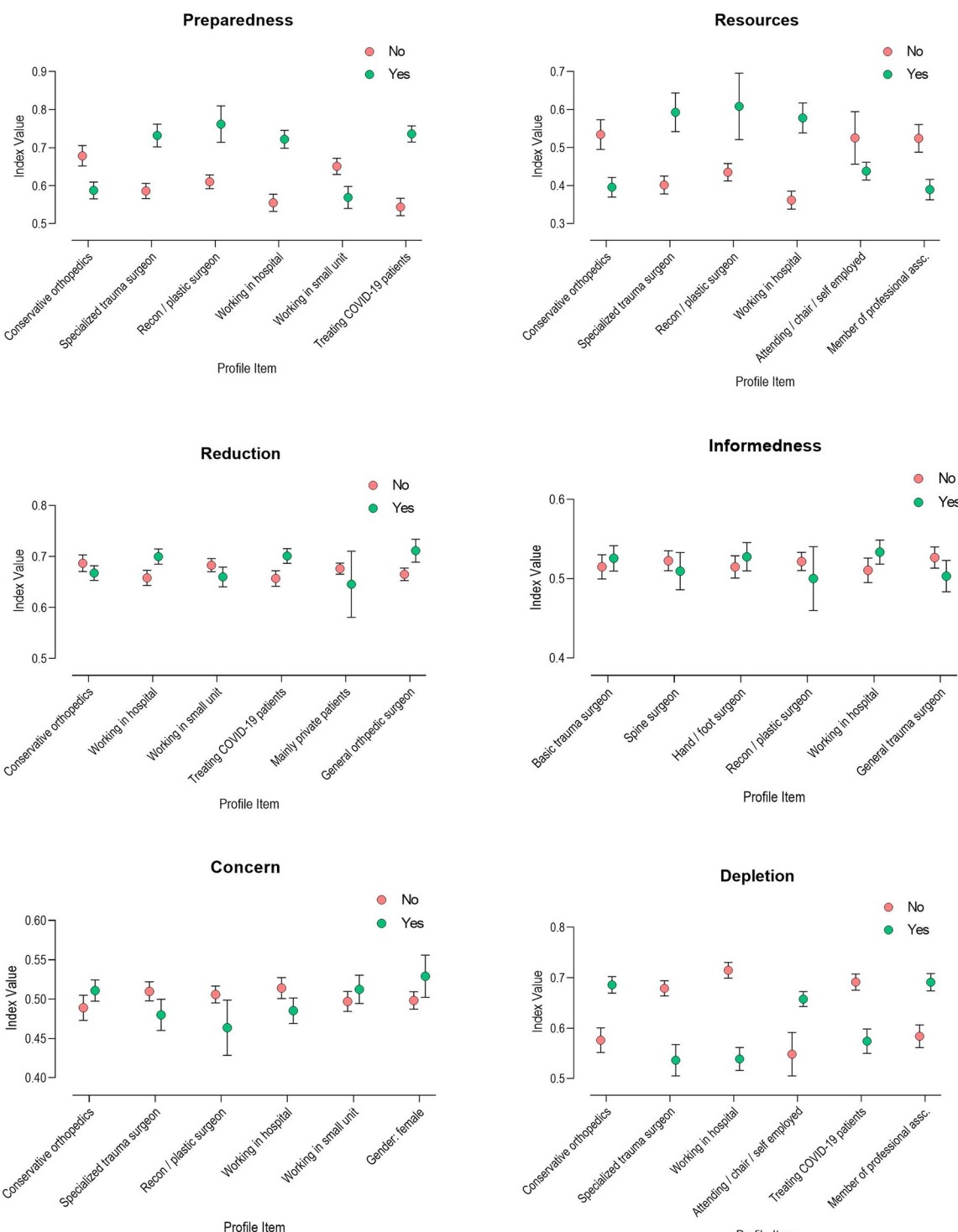

**Fig 4. Bivariate analysis of the profile items per index.** The graph shows the mean +/- 95% confidence interval of the index score for each of the six indices ("Preparedness": Fig 4A, "Resources": Fig 4B, "Informedness": Fig 4C, "Reduction": Fig 4D, "Concern": Fig 4E and "Depletion": Fig 4F), comparing subgroups that answered "yes" (green) or "no" (red) on those profile items that were most relevant / most significant in the bivariate analysis. For P-Values and significance levels please see Table 2.

**Table 3. Results of the multiple stepwise regression.**

| | IRR | δIRR |
|---|---|---|
| **Preparedness** | | |
| Treating COVID-19 patients | 1·234 | 0·234 |
| Working in hospital | 1·221 | 0·221 |
| Out-patient surgeon | 1·137 | 0·137 |
| Working in a small unit | 0·878 | -0·122 |
| Attending–Chairman–Selfemployed | 1·118 | 0·118 |
| Member of Professional Assoc. (BVOU) | 1·103 | 0·103 |
| Joint Surgeon | 1·078 | 0·078 |
| **Resources** | | |
| Working in hospital | 1·695 | 0·695 |
| Plastic / reconstructive surgeon | 1·195 | 0·195 |
| Treating COVID-19 patients | 0·861 | -0·139 |
| **Reduction** | | |
| Tumour surgeon | 0·921 | -0·079 |
| General orthopedic surgeon | 1·060 | 0·060 |
| Treating COVID-19 patients | 1·057 | 0·057 |
| Joint surgeon | 1·038 | 0·038 |
| **Informedness** | | |
| Working in hospital | 1·116 | 0·116 |
| General trauma surgeon | 0·919 | -0·081 |
| Spine surgeon | 0·925 | -0·075 |
| Hand / Foot surgeon | 1·051 | 0·051 |
| Age > 50 years | 1·048 | 0·048 |
| **Concern** | | |
| Gender: Female | 1·071 | 0·071 |
| Working in hospital | 0·951 | -0·049 |
| Basic trauma surgeon | 0·962 | -0·038 |
| **Depletion** | | |
| Working in hospital | 0·757 | -0·243 |
| Attending–Chairman–Selfemployed | 1·120 | 0·120 |
| General orthopedic surgeon | 1·066 | 0·065 |
| Pediatric orthopedic surgeon | 0·949 | -0·051 |

Shown are the profile items that were identified to be independent predictors of the indices, sorted by index and δIRR. The δIRR estimates the effect that a switch from "no" (0) to "yes" (1) of the particular item will have on the index in the regression model. E.g., working in a hospital will raise the index "Preparedness" by 0·221 points and lower the "Depletion" index by 0·243 points.

## Discussion

The COVID-19 pandemic is a global challenge for our society and healthcare system. We conducted this nationwide cross-sectional survey to identify the current demands and constraints of orthopedic and trauma surgery, both in clinic and in private practice. The results of our survey could provide useful conclusions for other nations but also for future crises. By addressing the survey to the members of the two largest German professional associations, a vast majority of all orthopedic and trauma surgeons was reached. The questionnaire was self-designed and not used or validated before, but the resulting data was homoscedastic and the Variance Inflation Factor (VIF) was low with 1.63. Symmetric and even distribution of the data, as well as

the excellent KMO criteria of 0.830 demonstrate that the sample is representative and suitable for the statistical analysis carried out.

Most of the orthopedic and trauma surgeons surveyed (71.0%) consider themselves well informed by the government and a clear majority considers the measures taken to be necessary (81.4%) and adequate (67.9%). This reflects the fact that our speciality, despite significant constraints of its own, considers the benefits for society to be important and strongly supports the measures. While the hospitals still report no severe shortages of material, shortages are reported in the supply of PPE by the participants. Only 26.5% and 23.7% of the respondents report sufficient stocks of masks or other PPE. This demonstrates the lack of preparation to a global pandemic that can be found worldwide. While in Germany an overload of hospital capacity could be prevented so far, the serious consequences of an overchallenged health care system were exhibited in a drastic fashion in northern Italy and Spain [18], but recently also in New York City. But while these developments are shocking, they are hardly unforeseen: In 2012, a report for risk analysis of civil protection of the German government, a pandemic of "Modi-SARS" was played out theoretically to assess the impact and identified a possible lack of preparation in advance. The calculated scene shockingly resembles the current situation and predicts a shortage of pharmaceuticals, medical devices, PPE and disinfecting agents.

In the multi-regression model, working in a hospital showed as an independent positive predictor for the indices "Preparedness" and "Resources". This is reasonable, since hospitals, compared to practices, have considerably more personnel, financial resources, equipment, and specialized institutions such as a pandemic task force. Being involved in COVID-19 patient care is also a positive predictor of "Preparedness", institutions are growing with the challenge. Redistribution of personnel, structural protective measures such as working in different teams and in separated areas for COVID-19 care, were started early in many institutions in Germany, and have apparently made enabled adequate preparation. Likewise, the presence of COIVD-19 patients was a negative predictor for "Resources", indicating constraints in the availability of PPE as mentioned above.

Being specialized in joint surgery is an independent positive predictive factor in the index "Reduction". This is clearly comprehensible, since from mid-March onwards, all elective operations in Germany had to be postponed, especially orthopedic operations such as arthroplasty. Working in a hospital is an independent positive predictor for the "Informedness" index. This may indicate an appropriate information policy of the hospital management or the pandemic task force. But it may also be related to the presence of different departments directly involved in COVID-19 treatment, providing sufficient information to their colleagues. Mind that the index only reflects the self-rated level of informedness, not the actual amount of knowledge present. Female surgeons appeared to be a predictor for a slightly raised level of concern. Gender differences regarding concern or fear are known and should be given more consideration in the future, especially in crisis situations [19]. Self-employment was an independent positive predictor in the index for financial concerns. "Depletion", while working in a hospital was a negative predictor. This points to the greater burden on self-employed surgeons, which should be taken into account when trying to provide support in a primary care setting.

The pandemic has led to massive restructuring in the healthcare system, with a substantial reduction in elective operations and outpatient department capacity and a sharp overall drop in patient numbers. Of the 20 most frequently performed operations in Germany, half are trauma surgery and orthopedic procedures. Of those, more than 50% are elective operations such as total joint arthroplasty, holding true for most industrial nations, e.g. the USA [20, 21]. This can be particularly threatening to the existence of surgeons with their own practice. 30.0% of participants assume that the pandemic and its consequences could threaten their existence. Overall, only 26.2% consider the compensation of the financial consequences to be

sufficient and 62.8% would appreciate more financial support from the ASHIP. The German healthcare system is among the best in the world, yet still underfunded [22]. The consequences will hit even harder if governmental or institutional health care plans are rare and poorly regulated. Despite the serious disruptions, the pandemic may also create opportunities, with 32.1% reporting more use of telemedicine procedures and home office and 43.8% expecting its value to rise in the future.

Despite the large number of participants and presumably a representative sample, distortions caused by the non-response bias must be considered. Due to the rapid pandemic development, prior pilot testing has not been performed and the lack of psychometric assessment should be borne in mind. Since we conducted a nationwide survey, the regional differences of the spread of COVID-19 may lead to distortions in the assessment of the pandemic due to areas affected to varying extents. When the survey was conducted, the COVID-19 pandemic was still in its early stages and expected to further progress. Our survey is a first impression of the impact this pandemic has on orthopedic and trauma surgery in Germany. We encourage its use among other nations and other specialties, to generate comparable results. In order to assess the overall effects of the pandemic more accurately, there will be follow-up surveys.

Essentially, being employed at the hospital proved to be a positive feature in the crisis; colleagues assumed that they were well prepared and informed, had sufficient resources and suffered less from financial concerns. Particular support should be given to self-employed surgeons in coping with the consequences of the COVID-19 pandemic. We could show that the index "concern" had a strong negative correlation with the indices "informedness" and "resources", so in future handling of the crisis, information and resources are the key factors to diminish healthcare professionals' anxiety. The small but significant gender gap in overall concern should be taken into closer consideration for future crisis management in order to be able to react appropriately.

As mentioned before and as foreseen by simulations, a massive lack of PPE has been reported particularly by the self-employed. Thus, the government should increase its efforts to stock up on PPE and consider this issue for future crises. Orthopedic and trauma surgeons in Germany advocate and support the measures taken to contain the COVID-19 pandemic. The preliminary success in the fight against the pandemic in Germany demonstrates that the appropriate measures have been implemented. The existence of effective healthcare structures, especially well-equipped hospitals in terms of personnel and funding, have proven to be indispensable in a national and global health crisis. This should not be forgotten in the future, when discussions about cost-cutting measures and restructuring in the healthcare systems will resurface in Germany and worldwide.

## Supporting information

**S1 Fig. Correlation matrix of index questions and indices.** The graph shows a heat map of the spearman correlation between those questionnaire items that were used to calculate the six indices (bold italics). The numbers refer to the item on the questionnaire. For a full list of the indices and the mode of calculation, see S1 Table. Red boxes indicate a negative correlation (Spearman r < 0), green boxes indicate a positive correlation (Spearman r > 0), with darker color representing stronger correlation and white or light colored boxes no or weak correlation. It can be seen how almost all questions assigned into one index correlate very well with the index itself, as well as with the other questions in that index (red frame boxes), as a method of validation for the indices. COVID-19: Coronavirus Disease 19; PPE: Personal protection equipment; PA: Professional association of orthopedics and trauma surgeons; ASHIP:

Association of statutory health physicians.
(JPG)

**S2 Fig. The bars show the mean +/- 95% confidence interval (error bars) of the six index values, comparing the subgroups of participants primarily working in a hospital (clinicians, grey bars) vs. those that do not work primarily work in a hospital (non-clinicians, white bars).** **** indicates an uncorrected P < 0.00038 and thereby significant difference between the groups with manual Bonferroni correction.
(JPG)

**S1 Table. Composition of the six indices from the questionnaire items and their weighting in the indices.**
(DOCX)

**S2 Table.**
(DOCX)

**S1 Data.**
(DOCX)

**S2 Data.**
(DOCX)

**S3 Data.**
(PDF)

## Acknowledgments

The authors warmly thank the German Society for Orthopedic and Trauma surgery (DGOU) and the Professional Association of Orthopedic and Trauma Surgery (BVOU) and in particular the Executive Board of both associations for their support. We acknowledge the helpful support the Department of Craniomaxillofacial Surgery, University Hospital Bonn, Germany and the Center for Development Research, University Hospital Bonn, Germany for their advice on and help with STATA and multiple regression models. The authors would like to thank the participants of the survey for their time and effort. We acknowledge support from the German Research Foundation (DFG) and the Open Access Publication Funds of Charité – Universitätsmedizin Berlin.

## Author Contributions

**Conceptualization:** Thomas M. Randau, Max Jaenisch, Adnan Kasapovic, Johannes Flechtenmacher, Matthias Pumberger.

**Data curation:** Thomas M. Randau, Max Jaenisch, Henryk Haffer, Friederike Schömig, Adnan Kasapovic, Katharina Olejniczak, Matthias Pumberger.

**Formal analysis:** Thomas M. Randau, Max Jaenisch, Friederike Schömig, Katharina Olejniczak.

**Methodology:** Thomas M. Randau, Katharina Olejniczak.

**Project administration:** Thomas M. Randau, Carsten Perka, Dieter C. Wirtz, Matthias Pumberger.

**Resources:** Johannes Flechtenmacher, Dieter C. Wirtz.

**Supervision:** Carsten Perka, Dieter C. Wirtz.

**Validation:** Henryk Haffer.

**Writing – original draft:** Thomas M. Randau, Max Jaenisch, Matthias Pumberger.

**Writing – review & editing:** Thomas M. Randau, Max Jaenisch, Henryk Haffer, Friederike Schömig, Adnan Kasapovic, Johannes Flechtenmacher, Carsten Perka, Dieter C. Wirtz, Matthias Pumberger.

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
