## [Decision Letter · Decision Letter 0]

3 Aug 2020

PONE-D-20-18877

Collateral Effect of COVID-19 on Orthopedic and Trauma Surgery

PLOS ONE

Dear Dr. Pumberger,

Thank you for submitting your manuscript to PLOS ONE. After careful consideration, we feel that it has merit but does not fully meet PLOS ONE’s publication criteria as it currently stands. Therefore, we invite you to submit a revised version of the manuscript that addresses the points raised during the review process.

The reviewers made an effort to help you to improve your manuscript. Please follow their recommendations.

We look forward to receiving your revised manuscript.

Kind regards,

Hans-Peter Simmen, M.D., Professor of Surgery

Academic Editor

PLOS ONE

Journal Requirements:

2. Thank you for stating in the text of your manuscript "The study protocol was reviewed and approved by the institutional ethics board (Approval No. #20/127)". Please also add this information to your ethics statement in the online submission form. In addition, in your ethics statement please state that this was an online anonymous survey.

3. Please include a summary table of any collected demographic data.

5.Thank you for stating the following in the Declaration of competing interests Section of your manuscript:

[P. C. reports personal fees from Smith&Nephew, personal fees from Link, personal fees from

DePuy/Synthes, personal fees from Aesculap, personal fees from Zimmer, outside the submitted work.

W. D. C. has nothing to disclose.]

 [The author(s) received no specific funding for this work.]

Additionally, because some of your funding information pertains to commercial funding, we ask you to provide an updated Competing Interests statement, declaring all sources of commercial funding.

In your Competing Interests statement, please confirm that your commercial funding does not alter your adherence to PLOS ONE Editorial policies and criteria by including the following statement: "This does not alter our adherence to PLOS ONE policies on sharing data and materials.” as detailed online in our guide for authors  http://journals.plos.org/plosone/s/competing-interests.  If this statement is not true and your adherence to PLOS policies on sharing data and materials is altered, please explain how.

Please include the updated Competing Interests Statement and Funding Statement in your cover letter. We will change the online submission form on your behalf.

Reviewers' comments:

Reviewer's Responses to Questions

**Comments to the Author**

1. Is the manuscript technically sound, and do the data support the conclusions?

Reviewer #1: Yes

Reviewer #2: Yes

2. Has the statistical analysis been performed appropriately and rigorously? 

Reviewer #1: Yes

Reviewer #2: Yes

3. Have the authors made all data underlying the findings in their manuscript fully available?

Reviewer #1: Yes

Reviewer #2: No

4. Is the manuscript presented in an intelligible fashion and written in standard English?

Reviewer #1: Yes

Reviewer #2: Yes

5. Review Comments to the Author

Reviewer #1: Dear authors,

thank you for your interesting survey on the impact of COVID-19 on the german orthopaedic and trauma surgery.

The development of the COVID-19 pandemia is nowadays well known and spread througout the media worldwide. Due to this background i would suggest to shorten the Introduction and focus on the development within germany as the base of your survey.

The description of Table 1 is not fully conclusive, please reformulate clearer: "Descriptive analysis and data quality, as assessed by N, mean, standard deviation (SD), standard error of the mean (SEM), 95% confidence interval (95% CI), skewness, and kurtosis of all items where these analyses was sensibly possible."

Reading your manuscript i was wondering if there is a positive correlation between the level of preparedness, resources, informedness and existing concerns. A further analysis of this context might lead to a better flow of information limiting concerns in the future.

Reviewer #2: The authors present results of a questionnaire based survey among trauma and orthopaedic surgeons in private practices and hospitals in Germany regarding the impact of COVID-19.

Abstract:

- Please provide full text before using the abbreviation “PPE”

Introduction:

- Well done.

Methods:

- As only about half of the opened questionnaires have been returned, it would be interesting to know whether the sample analyzed was representative. Do you have demographic data of the unreturned questionnaires or could you estimate these numbers based on numbers of hospital and outpatient trauma/orthopaedic surgeons in Germany?

- Supplementary Material (A) and (B) with the original questionnaire are missing.

- Whereas the index questions are clearly described in the methods part, the different groups of surgeons (independent profile variables) are not mentioned. Please add this to the methods part. Partially the information can be found in Figure 2. However, it is not congruent to the variables used in Table 2.

Results:

- It would be informative to have a “Table 1“ giving crude numbers of baseline data for the different independent profile variables (hospital employed, working in a small unit etc) and each index question.

- I am not sure whether the many numbers in Table 1 are easy to understand for the reader. It would be helpful to have an example for at least one predictor explained in text. E.g. “Reduction in outpatient clinic with a mean of 0.7673 means that …”

- In general, the manuscript is methodologically sound, but very “technical” and therefore not easy to understand for the reader. I suggest to add legends for Figures 3 and Suppl. Figure 1, each containing an example that explains how to read the diagrams.

Discussion:

- Well done.

6. PLOS authors have the option to publish the peer review history of their article (what does this mean?). If published, this will include your full peer review and any attached files.

Reviewer #1: **Yes: **Carina Eva Maria Pothmann, MD

Reviewer #2: No

---

## [Author Response · Author response to Decision Letter 0]

21 Aug 2020

We have addressed all comment of the editorial office and the reviewers, and hereby submit our point-by-point reply: 

We have checked the style requirements and updated the style of the manuscript as needed. 

2. Thank you for stating in the text of your manuscript "The study protocol was reviewed and approved by the institutional ethics board (Approval No. #20/127)". Please also add this information to your ethics statement in the online submission form. In addition, in your ethics statement please state that this was an online anonymous survey.

We have completed the ethics statement in the online submission as required 

3. Please include a summary table of any collected demographic data.

No other demographic data than those already displayed in Figure 2 was collected. We therefore do not see any additional value in also providing this data as a table. However, if needed, the data can be included, we have uploaded it as supplemental Table 2 in the online submission system. 

We have included figure legends to figures 1-4 in the manuscript.

5. Please remove any funding-related text from the manuscript 

Done

5. ..let us know how you would like to update your Funding Statement. Currently, your Funding Statement reads as follows:

 [The author(s) received no specific funding for this work.]

Still true. No adjustments necessary. 

Additionally, because some of your funding information pertains to commercial funding, we ask you to provide an updated Competing Interests statement, declaring all sources of commercial funding. Please include the updated Competing Interests Statement and Funding Statement in your cover letter. We will change the online submission form on your behalf.

None of the funding received by individual authors pertains any competing interest related to the study. No commercial funding was received in relation to the study. We have deleted the competing interest section within the manuscript. 

The development of the COVID-19 pandemia is nowadays well known and spread througout the media worldwide. Due to this background i would suggest to shorten the Introduction and focus on the development within germany as the base of your survey.

Only the first paragraph of 123 words relates to the pandemia in general, the rest of the introduction is already focused on the german development. We have further shortened the introduction to address the reviewers concerns. 

The description of Table 1 is not fully conclusive, please reformulate clearer: "Descriptive analysis and data quality, as assessed by N, mean, standard deviation (SD), standard error of the mean (SEM), 95% confidence interval (95% CI), skewness, and kurtosis of all items where these analyses was sensibly possible."

We have modified Table 1 description to be clearer in its’ meaning: “SD, SEM and 95% CI was calculated only for those items measured on the 5-degree scale or Likert skale, as well as for the calculated indices. For dichotome or multiselect items, these calculations are not sensibly possible”. 

Reading your manuscript i was wondering if there is a positive correlation between the level of preparedness, resources, informedness and existing concerns. A further analysis of this context might lead to a better flow of information limiting concerns in the future.

Figure 3 holds the information the reviewer is asking for, though hard to recognize. The lower right corner of the figure presents a correlation matrix between the calculated indices. It shows that “concern” (last line but one in the matrix) correlates strongly negative with “resources” and “informedness”, and positively with “depletion”, less positively also with “reduction”. On Page 10, “Correlations and regression”, we are pointing this out in the text. However, we agree that this point is worth pointing out in regard to future handling, and therefore now have picked this up in the discussions section: “We could show that the index “concern” had a strong negative correlation with the indices “informedness” and “resources”, so in future handling of the crisis, information and resources are the key factors to diminish healthcare professionals’ anxiety.” 

Please provide full text before using the abbreviation “PPE”

The abbreviation was defined inthe abstract, as requested.

As only about half of the opened questionnaires have been returned, it would be interesting to know whether the sample analyzed was representative. Do you have demographic data of the unreturned questionnaires or could you estimate these numbers based on numbers of hospital and outpatient trauma/orthopaedic surgeons in Germany?

We have completed an analysis of the incomplete surveys, and have attached this to the submitted revision. The PDF “SupplData3_ Sample Size and selction bias” addresses the reviewers very good question. If needed, the data can also be included in the final manuscript as supplementary material. 

Supplementary Material (A) and (B) with the original questionnaire are missing.

We have included the original and the translated questionnaire (originally referenced as “A”) as supplementary data 1 and 2 in the resubmission, as well as the table of index questions (originally “B”) as SupplTable1_Index questions. 

Whereas the index questions are clearly described in the methods part, the different groups of surgeons (independent profile variables) are not mentioned. Please add this to the methods part. Partially the information can be found in Figure 2. However, it is not congruent to the variables used in Table 2.

With appending the original survey, the different groups of surgeons and professional levels are clearly defined. We see this as sufficient for the reader, and have added the original and translated survey as intended. Table 2 only lists those independent profile variables that were significant in the bivariate analysis, and is therefore not congruent with the figure 2, which is a mere summary of the answers given. 

It would be informative to have a “Table 1“ giving crude numbers of baseline data for the different independent profile variables (hospital employed, working in a small unit etc) and each index question.

In this case we do not agree with the reviewer. Figure 1 and 2 are a graphical representation of the numbers, as the answers were given by the participants, an additional table with the numbers would not add any information to this. Should the editor decide that a table with numbers is beneficial rather than only figures, we can provide this. 

I am not sure whether the many numbers in Table 1 are easy to understand for the reader. It would be helpful to have an example for at least one predictor explained in text. E.g. “Reduction in outpatient clinic with a mean of 0.7673 means that …”

We have added a paragraph to the results section, explaining how Likert scales transform into numerical values to be statistically analyzed, and how to grade this. We abstain from explaining Skewness, Kurtosis and other statistical measures, assuming the statistically inclined reader will be familiar and able to interpret. 

In general, the manuscript is methodologically sound, but very “technical” and therefore not easy to understand for the reader. I suggest to add legends for Figures 3 and Suppl. Figure 1, each containing an example that explains how to read the diagrams.

We have added figure legends (see above), and included an example as suggested in Figure legend 3. 

We thereby see all issues raised by the reviewers as addressed and would very much appreciate our manuscript to be considered for publication.

---

## [Editor Report · Decision Letter 1]

25 Aug 2020

Collateral Effect of COVID-19 on Orthopedic and Trauma Surgery

PONE-D-20-18877R1

Dear Dr. Pumberger,

We’re pleased to inform you that your manuscript has been judged scientifically suitable for publication and will be formally accepted for publication once it meets all outstanding technical requirements.

Kind regards,

Hans-Peter Simmen, M.D., Professor of Surgery

Academic Editor

PLOS ONE
---

## [Editor Report · Acceptance letter]

27 Aug 2020

PONE-D-20-18877R1 

Collateral Effect of COVID-19 on Orthopedic and Trauma Surgery 

Dear Dr. Pumberger:

I'm pleased to inform you that your manuscript has been deemed suitable for publication in PLOS ONE. Congratulations! Your manuscript is now with our production department. 

Kind regards, 

on behalf of

Dr. Hans-Peter Simmen 

Academic Editor

PLOS ONE